# Dietary Walnuts Prevented Indomethacin-Induced Gastric Damage via AP-1 Transcribed 15-PGDH, Nrf2-Mediated HO-1, and n-3 PUFA-Derived Resolvin E1

**DOI:** 10.3390/ijms25137239

**Published:** 2024-06-30

**Authors:** Jong Min Park, Ki Baik Hahm

**Affiliations:** 1College of Oriental Medicine, Daejeon University, Daejeon 34520, Republic of Korea; 2CHA Cancer Preventive Research Center, CHA Bio Complex, Seongnam 13488, Republic of Korea

**Keywords:** indomethacin, gastric damage, walnut, 15-PGDH, Nrf2

## Abstract

Non-steroidal anti-inflammatory drugs (NSAIDs), the most highly prescribed drugs in the world for the treatment of pain, inflammation, and fever, cause gastric mucosal damage, including ulcers, directly or indirectly, by which the development of GI-safer (-sparing) NSAIDs relates to unmet medical needs. This study aimed to document the preventive effects of walnut polyphenol extracts (WPEs) against NSAID-induced gastric damage along with the molecular mechanisms. RGM-1 gastric mucosal cells were administered with indomethacin, and the expressions of the inflammatory mediators between indomethacin alone or a combination with WPEs were compared. The expressions of the inflammatory mediators, including COX-1 and COX-2, prostaglandin E^2^, 15-hydroxyprostaglandin dehydrogenase (15-PGDH), and antioxidant capacity, were analyzed by Western blot analysis, RT-PCR, and ELISA, respectively. HO-1, Nrf-2, and keap1 were investigated. The in vivo animal models were followed with in vitro investigations. The NSAIDs increased the expression of COX-2 and decreased COX-1 and 15-PGDH, but the WPEs significantly attenuated the NSAID-induced COX-2 expression. Interestingly, the WPEs induced the expression of 15-PGDH. By using the deletion constructs of the 15-PGDH promoter, we found that c-*Jun* is the most essential determinant of the WPE-induced up-regulation of 15-PGDH expression. We confirmed that the knockdown of c-*Jun* abolished the ability of the WPEs to up-regulate the 15-PGDH expression. In addition, the WPEs significantly increased the HO-1 expression. The WPEs increased the nuclear translocation of Nrf2 by Keap-1 degradation, and silencing Nrf2 markedly reduced the WPE-induced HO-1 expression. We found that the WPE-induced HO-1 up-regulation was attenuated in the cells harboring the mutant Keap1, in which the cysteine 151 residue was replaced by serine. These in vitro findings were exactly validated in indomethacin-induced gastric rat models. Daily walnut intake can be a promising nutritional supplement providing potent anti-inflammatory, antioxidative, and mucosa-protective effects against NSAID-induced GI damage.

## 1. Introduction

Patients who take non-steroidal anti-inflammatory drugs (NSAIDs) have an increased risk of the ulceration of the gastrointestinal mucosa, from erosions to ulcers, although it is less common than *Helicobacter pylori* or other ulcers of the stomach or duodenum, which can lead to life-threatening complications [1,2,3]. Therefore, the novel strategies to address long-term NSAID-associated gastrointestinal damage are essential. In addition to the current drugs based on the classic damaging mechanisms attributable to the decline in gastric mucosal prostaglandin (PG) synthesis, reductions in the mucosal blood flow, attenuated bicarbonate secretion, and mucus synthesis related with PG levels [4,5], more effective therapeutics targeting the update mechanisms of NSAID-induced gastrointestinal (GI) damage are introduced or under clinical trials based on the recent advances in basic science and biotechnology exploring the deeper molecular mechanisms beyond COX inhibition, e.g., *Coxibs*, NO, or H2S-releasing NSAIDs, NSAIDs inhibiting autophages, and combinations with cytoprotective agents [6,7,8,9,10]. In addition, phytochemicals, such as dietary phenolic compounds, phenolic acids, flavonols, flavonoids, gingerols, carotenes, and organosulfur, are under investigation in order to address the NSAID-induced GI damage via the antioxidant, anti-inflammatory, and antibacterial benefits of those natural agents.

Although the ulcerogenic mechanism of indomethacin is still unclear, it induces gastric damage via inhibiting cyclooxygenase-1 (COX-1), whereas increasing PGE_2_ decreases bicarbonate and mucus, increases the oxidant stress, and decreases the antioxidant parameters [11]. Therefore, designing GI-safer (or sparing) NSAIDs [12,13] is undertaken based on these fundamental mechanisms of GI damage, including *coxibs* as a selective inhibitor of COX-2 [14], NO-NSAIDs [15,16], phosphatidylcholine-NSAID [17], and hydrogen sulfide-releasing NSAID [18].

Supported by our previous publications that *fat*-1 transgenic mice producing n-3 PUFAs in the gastrointestinal tract by the overexpression of 3-*desaturase* significantly ameliorated the indomethacin-induced GI damage [19,20], in this study, we assert the hypothesis that walnut polyphenol extracts (WPEs) containing n-3 PUFAs can be a food factor to address the NSAID-induced GI damage because of plentiful n-3 PUFAs and the feasibility of the food factor. In this investigation, we explore the efficacy of WPEs against indomethacin-induced gastric damage in RGM-1 gastric cells and in vivo animal models. With novel mechanisms such as 15-PGDH induction via AP-1, HO-1 increases via Nrf2 induction, and resolvin E1 (RvE1), the possibility of dietary walnut extracts for ameliorating NSAID-induced damage is presented.

## 2. Results

### 2.1. WPE-Inhibited PGE_2_ via COX-2 and NF-κB Inhibition

With indomethacin administration, the gastric cells showed significant increases in cox-2 mRNA expression (Figure 1A). However, in the presence of the walnut extracts, the expression of cox-2 mRNA was significantly decreased. These results were noted in the consequent Western blot of COX-2 (Figure 1B). Interestingly, in this experiment, although COX-1 has been known as constitutive, as shown in Figure 1A,B, the WPEs increased either the *COX-1* mRNA or its expression, signifying that the WPEs significantly regulated the expression of COX. These changes in COX led to significant increases in PGE_2_ with indomethacin administration. However, the WPEs significantly decreased the PGE_2_ levels. All these changes were significantly correlated with the repression of NF-κB and significant decreases in NF-κB nuclear translocation (Figure 1D).

### 2.2. Walnut Extracts Prevented Indomethacin-Induced Inhibition of 15-PGDH

As noted in the COX regulation with WPEs (Figure 1) that the changes in the COX expression and PGE_2_ decrease, we proceeded to measure the changes in 15-PGDH. Indeed, 15-PGDH has been known to reflect the PG levels. As shown in Figure 2A, indomethacin alone significantly decreased in the *15-PGDH* mRNA, but, in the presence of the WPEs, the cancelled expression of the 15-PGDH mRNA was significantly restored. These findings in the 15-PGDH mRNA were repeated with the Western blot for 15-PGDH. As noted in Figure 2B, the indomethacin administration significantly decreased 15-PGDH, but not in the presence of the WPEs. Overall, the WPEs significantly afforded anti-inflammatory action through either COX-2 inhibition, COX-1 increases, or the preservation of 15-PGDH. In order to know the promoter regulation of 15-PGDH with WPEs, we conducted a promoter assay. As shown in Figure 2C, a promoter containing -1024 showed significant increases in promoter activity in the presence of the WPEs (*p* < 0.001). Looking at -1024 to 1 of 15-PGDH, there was one *Ets* and three binding sites of AP-1. Therefore, when we measured the c-*Jun* expression with walnut, the expression of c-*Jun* was significantly increased as time passed after the WPE administration (Figure 2D), the findings of which were further confirmed by the confocal imaging of c-*Jun* (Figure 2E) or AP-1 luciferase activities (Figure 2F), consistent with the findings of the Western blot of c-*Jun*. When we knocked down c-*Jun* with siRNA, as shown in Figure 2G, in spite of the WPE administration, 15-PGDH was not increased.

### 2.3. Walnut Extracts Afforded Antioxidative Action under Indomethacin Stimulation

In order to verify the antioxidative capacity of the WPEs, we performed a DPPH reduction assay. As shown in Figure 3A, the antioxidative actions of the WPEs were shown from the concentration higher than 5 μg/mL, which was verified with a flow cytometry analysis (*p* < 0.01, Figure 3B). Among the antioxidative enzymes, the WPEs exerted significantly increased expressions of HO-1, PRX2, and GPX2 (*p* < 0.01, Figure 3C). The significantly increased expression of NOX-1 with indomethacin was significantly attenuated with the WPE administration (Figure 3D). In terms of the RT-PCR for *HO-1* and several *HSP* mRNAs, the walnut increased the *HO-1* mRNA as well as *HSP70* and *HSP27* mRNAs, after which we measured the HO-1 luciferase activities. As shown in Figure 3D, the WPEs significantly increased HO-1 luciferase (*p* < 0.01). Conclusively, walnut, although not a professional antioxidant, induced significant cytoprotection against indomethacin via antioxidative actions.

### 2.4. Significant Induction of Nrf2 via keap1 Inhibition Led to HO-1 Induction

Since HO-1 is a significant antioxidative gene and walnut rescued the stomach from indomethacin via HO-1, we established the hypothesis that walnut can activate Nrf-2 to induce HO-1. As shown in Figure 4A, Nrf2 was significantly increased with the increasing time of the 5 μM WPE. In a similar setting, we measured the antioxidant response element (ARE) luciferase activities and found that walnut significantly increased the ARE activities (*p* < 0.01, Figure 4B). Since Nrf2 is activated after inactivating keap1, we also measured the expression of keap1 with an increasing dose of WPE. As noted in Figure 4C, the significantly decreased expression of Nrf2-inactivated Keap1 (*p* < 0.01). The findings included that, when Nrf2 was knocked down, the walnut administration did not provoke HO-1 (Figure 4D), and also, when keap1 was knocked down, the WPE administration did not result in HO-1 induction (Figure 4E), which consistently signified that the antioxidative and cytoprotective actions of the WPEs were through HO-1 transcribed with Nrf2.

### 2.5. Anti-Apoptotic Action of Walnut Extracts against Indomethacin Cytotoxicity

Indomethacin alone led to significant cytotoxicity assessed via the MTT assay (Figure 5A), but the co-administration of indomethacin and WPEs at higher than 5 mg/mL concentrations significantly ameliorated the indomethacin cytotoxicity. These MTT results were further verified with flow cytometry (Figure 5B). As an explanation regarding how walnut ameliorated the indomethacin-induced cytotoxicity, a significantly increased expression of Bcl-2 was found (*p* < 0.001, Figure 5C).

### 2.6. RvE1 Induction with WPE to Increase Cytoprotection

Since the features of the anti-inflammatory and cytoprotective actions of n-3 fatty acid-containing substances are through the increased resolving action of inflammation, under the hypothesis that walnuts plentiful in n-3 polyunsaturated fatty acids (PUFAs) can increase RvE1, we measured RvD1 and RvE1, respectively. As shown in Figure 6B, significantly increased levels of RvE1 were noted in accordance with an increasing dose of the WPE even under the indomethacin challenge. However, no significant changes were noted in RvD1 (Figure 6A).

### 2.7. Dietary Walnut Led to Significant Rescue from Indomethacin-Induced Gastric Damage

Stimulated by the above in vitro results that walnut afforded a significant improvement regarding indomethacin-induced cytotoxicity, we checked the in vivo actions of walnut against indomethacin-induced gastric damage (Figure 7A). The co-administration of indomethacin and walnuts (50, 100, and 200 mg/kg) significantly decreased the gross lesion index (*p* < 0.05, Figure 7A). Upon a pathological analysis, significant erosive and ulcerative changes were noted in the indomethacin-administered group, but these muco-destructive changes as well as inflammation were significantly ameliorated in the walnut co-treated group (*p* < 0.05, Figure 7B). Using mucosal homogenates, COX-2, COX-1 (Figure 7C), 15-PGDH (Figure 7D), NF-κB, STAT3, c-*Jun* (Figure 7E), HO-1, and Nrf2 (Figure 7F) were measured, respectively. Consistent with the above in vitro experiment, significant decreases in COX-2, induction of COX-1 and 15-PGDH, c-*Jun* nuclear translocation for acute phase response, decreased NF-κB, and significant induction of HO-1 via Nrf2 were noted in the group treated with a walnut-containing diet.

## 3. Discussion

Ameliorating NSAID-induced GI damage is quite an urgent unmet medical need because of the high mortality and increasing prevalence due to the increasing life-expectancy accompanied with high prescriptions of NSAIDs. Either the GI-safe NSAID development or co-administration of NSAIDs with agents providing cytoprotection might be an ideal strategy [21,22]. In this study, the authors effectively concluded that the dietary intake of fresh walnut can be a convincing way to avoid NSAID-induced GI damage, achieve a secured mode of action, and establish a proof of concept. Considering that developing GI-safer NSAIDs is one of the most urgent unmet medical needs, our current study is quite novel and promising because the dietary intake of walnuts can accomplish the unmet medical needs beyond GI-safer NSAIDs or the enhancement of NSAIDs’ effects under safety [23,24]. As summarized in Figure 8, the co-administration of NSAIDs and WPEs can cover the untoward mechanisms of GI cytotoxicity and enforce cytoprotection. However, during the current experimental procedure (see the Appendix A), we found that higher amounts of walnut are not always beneficial, and the walnuts were very vulnerable to peroxidation. In terms of the clinical benefit, we inferred that a vacuum package to prevent lipid peroxidation and optimal dosing should be considered (see Appendix A). As noted in Figure 1, the walnut extract doses were quite important, and the dose should be determined according to the hidden mode of action, Nrf2 activation, Keap1 inaction, and HO-1 induction, and the 15-PGDH induction should also be factored in.

Although the basic mode of action of NSAIDs is the inhibition of COX to reduce the PGs responsible for either inflammation or pain, GI damage also occurs paradoxically via COX induction [25,26,27]. In this condition, 15-PGDHs catabolize PGE_2_, simply a COX-2 antagonist, reducing the inflammatory and mutagenic PGs and rendering 15-PGDH-tumor-suppressive and inflammation-reducing results [28,29,30,31]. As shown in Figure 2A, the indomethacin administration resulted in significant reductions in the *15-PGDH* mRNA after 24 h, but the walnut co-administration significantly induced *15-PGDH* mRNA and protein expression (Figure 2B), thus identifying for the first time that walnut can induce 15-PGDH, confirmed with the promoter assay of 15-PGDH (Figure 2C). Since the biological significance of 15-PGDH induction is either tumor-suppressive or inhibits inflammation, walnut administration seems to immediately operate 15-PGDH via an AP-1 early response element. Therefore, translated with the results shown in Appendix A, the co-administration or intake within less than 1h of NSAIDs seems to be important to benefit the 15-PGDH induction in terms of combatting NSAID cytotoxicity.

NSAIDs, including aspirin, indomethacin, diclofenac, and ketoprofen, are widely used in clinical medicine, but these drugs may cause oxidative stress, leading to GI damage such as ulcers [32]. Therefore, the reduction in oxidative stress may be an effective curative strategy for preventing and treating the NSAID-induced ulcers of the GI mucosa. As shown in Figure 3A,B,D, indomethacin administration led to significant increases in oxidative stress, as assessed with the DPPH reduction assay, DCF-DA flow cytometry, and NOX expression. However, the WPEs afforded significant induction regarding either the antioxidative or cytoprotective genes, including HO-1, HSPs, and GPX2. Especially, the HO-1 induction via Nrf2 was prominent. Since neutrophils- or reactive oxygen species (ROS)-dependent microvessel injuries occur because of the ROS produced by the activated neutrophils after indomethacin damage, the cell membrane, cytoplasmic protein, and even nuclear DNA are prime aspects in indomethacin-induced GI damage [33], and diverse kinds of formulas have been investigated. For instance, the GI protection by quercetin against indomethacin-induced oxidative stress and inflammation via inhibiting NF-κB, NOX, and ICAM-1 [34], eupatilin from the ethanol extracts of *Artemisia asiatica*-mediated protective action of epithelial cells against indomethacin via both the ERKs and PI3K/Akt pathways, as well as Nrf2 translocation [35] and the Nrf2-mediated HO-1 induction of PPIs afforded significant protective effects against NSAID-induced gastric damage beyond acid suppressive actions [36], and rebamipide exerted HO-1 induction under indomethacin damage [37].

Heme oxygenase-1 (HO-1), the rate-limiting enzyme in the catabolism of heme, followed by the production of biliverdin, free iron, and carbon monoxide (CO), is a stress-responsive protein induced by various oxidative agents [38,39]. While HO-1 activity is associated with obesity, metabolic syndrome, and even cardiovascular diseases, in the GI tract, HO-1 is shown to be transcriptionally implicated in ischemia-reperfusion, indomethacin-induced damage, lipopolysaccharide-associated sepsis, pancreatitis, and inflammatory bowel disease, indicating that the activation of HO-1 may act as an endogenous defensive mechanism to reduce the inflammation and tissue injury in the GI tract [37,40,41].

In the GI tract, since the anti-apoptotic response of the WPEs was not so prominent in protecting against NSAIDs, we established the hypothesis that the host-driven resolving mechanism, such as defensin or resolvin, might be stimulated with WPEs; resolvinE1 (RvE1), as an endogenous lipid mediator derived from n-3 PUFAs, contributes to the resolution of allergic inflammatory responses [42]. As a result, as shown in Figure 6B, RvE1 significantly increased as the WPE dose increased, by which we could conclude that the antioxidant and anti-inflammatory actions of the WPEs are the main mechanisms involved in the gastroprotective effects against indomethacin-induced gastric damage via RvE1. Although not documented in the current study, since RvE1 receptor ChemR23 is usually expressed in intestinal epithelial cells [43], we speculated that the DHA or EPA in WPEs were stimulated to afford an RvE1-mediated inflammatory resolution as well as anti-apoptotic cytoprotection.

These molecular mechanisms drawn from the in vitro cellular model of NSAID-associated cell damage in terms of NSAID-induced GI damage were validated in an in vivo model of NSAID-induced gastric damage involving 50 mg/kg indomethacin administered orally and eliminated 16 h later (Figure 7). The gastric damage was significantly ameliorated with 50, 100, and 200 mg/kg walnut-containing pellet diets (*p* < 0.01) assessed grossly and histologically. The COX-2 inhibiting action, NF-κB repressing action, and STAT3 inactivating action were all significantly operational in the group treated with dietary walnut (*p* < 0.01), while the HO-1-inducing, Nrf2-activating, and 15-PGDH-preserving mechanisms were all significantly in concert with the dietary walnut (*p* < 0.01) against indomethacin-associated gastric damage. Furthermore, the anti-apoptotic executions with WPEs contributed to the rescuing action against NSAID cytotoxicity. The increasing apoptotic cell death has been revealed as the contributing cytotoxic mechanism of NSAIDs-associated damage (Figure 5 and Figure 7).

Walnuts (*Juglans regia* L.) are rich in the essential fatty acids, vitamin E, and folate. Especially, WPEs contain high concentrations of α-linoleic acid, α-linolenic, oleic acid, and γ-tocopherol. They also contain large amounts of polyphenols, including ellagic acid, gallic acid, and quercetin [44]. In the metabolite-profiling analysis, walnut caused a significant increase in several polyunsaturated fatty acids (PUFAs), including docosahexaenoic acid (DHA) and 9-oxo-10(E),12(E)-octadecadienoic acid (9-oxoODA), as well as kynurenic acid. Walnuts exhibit a wide range of health benefits, including anti-inflammatory, antioxidative, and regenerating actions. For instance, dietary walnut supplementation showed significant effects in terms of the recovery from dextran sulfate sodium-induced colitis [45,46] and significant protection against fenitrothion- or malathion-mediated immunotoxicity [47], reducing the telomere length [48], ameliorating colitis and colitis-associated cancer [46], suppressing colon cancer cell growth, and regulating anti-cancer stem cells [49]. Although not covered in the current investigation, cancer stem cell markers including CD133, CD44, DLK1, and Notch1 as well as the β-catenin/p-GSK3β signaling pathway were significantly down-regulated, and the self-renewal capacity of CSCs was suppressed. Taken together with our investigation, WPEs can impose significant rescuing action against NSAID-induced GI damage.

From the current investigation, as secure NSAID safety, we considered the possibility of dietary intervention in elderly populations taking long-term NSAIDs due to neurodegenerative diseases or the general population taking NSAIDs, such as in terms of desmoids tumors, juvenile arthritis, and more, in order to prevent NSAID-associated GI damage using a dietary co-administration with NSAIDs and walnuts. However, as shown in Appendix A, the walnuts should be fresh (the WPE prepared with the 1 h method should be used because the WPE prepared and stored for 48 h aggravated the oxidative stress; Appendix A), the dose should be determined (the WPE administered at 5 μg/mL prepared 1h before was effective in reducing the oxidative stress, while 20–40 μg/mL increased the oxidative stress; Appendix A), and the hermetic principle should be considered because higher doses of WPEs (>160 μg/mL) showed significant cytotoxicity (*p* < 0.01, Appendix A). Lastly, we are curious as to whether the whole walnut polyphenol extract (WPE) or the individual polyphenol components of walnut, including oleic acid (Appendix A) [50], catechin (Appendix A) [44,51,52], and ellagic acid (Appendix A) [53,54], impose similar rescuing actions against NSAIDs. As result, our WPE seems to be the best as it pertains to each tested component of the walnut when compared to the DCF-DA inhibiting action under 500 μM and 1h indomethacin administration. However, a detailed clinical trial should be conducted in order to establish walnuts as a dietary intervention to secure the GI safety of NSAIDs.

## 4. Materials and Methods

### 4.1. Reagents

Indomethacin (IND) as NSAID was purchased from Sigma Aldrich (St. Louis, MO, USA). Primers for RT-PCR were synthesized by Macrogen (Seoul, Republic of Korea). Antibodies were purchased from Cell Signaling Technology (Beverly, MA, USA) and Santa Cruz Biotechnology (Santa Cruz, CA, USA). Horeseradish peroxidase-conjugated anti-mouse/rabbit/goat IgG was purchased from Santa Cruz Biotechnology (Santa Cruz, CA, USA).

### 4.2. Preparation of WPE

Walnut polyphenol extract (WPE) from English walnuts (*J. regia*, California Walnut Commission) was prepared according to a previously described methanolic extraction method [55]. Briefly, after the walnuts were frozen for 24 h, the shelled kernels were finely ground and immersed in a solution of 75% acetone containing 526 µm/L sodium metabisulfite. The solution was subsequently purged with N2 to prevent oxidation and was incubated at 4 °C. After 24 h, the solution was decanted, thereby resulting in a cold extract that was centrifuged at 8000× *g* for 10 min. The resulting supernatant was filtered using Whatman filter paper No. 2. To remove lipids from the sample, the acetone was removed under reduced pressure, and methanol (50% aqueous, *v*/*v*) was added. After three consecutive hexane extractions, the extracts were lyophilized to a dry powder after removing the methanol to prevent oxidation. All the prepared samples were stored at 80 °C until needed.

### 4.3. Cells and Cytotoxicity Assay

The rat gastric mucosal cells, RGM1, were maintained at 37 °C in a humidified atmosphere containing 5% CO_2_ and cultured in Dulbecco’s modified Eagle’s medium containing 10% (*v*/*v*) fetal bovine serum and 100 U/mL penicillin. Cells were treated with IND and WPEs in DMSO and used at the final concentrations indicated in the text and in figure legends. Cells were preincubated with various concentrations of WPEs for 6 h and treated with 500 μM IND for the time indicated in the text and in figure. Cell cytotoxicity was measured by MTT, [3-(4,5-dimethylthiazol-2-yl)-2,5-diphenyltetrazolium bromide], assay.

### 4.4. Dichlorofluorescin Diacetate (DCF-DA) Measurement

For detecting the accumulation of ROS in RGM1, cells were monitored using the fluorescence-generating probe DCF-DA. Cells were rinsed with HBSS solution and loaded with 10 μM DCF-DA. After 30 min incubation at 37 °C, cells were analyzed with flow cytometry.

### 4.5. Animal Experimental Procedure

The C57BL/6mice were purchased from Orient bio (Seoul, Republic of Korea). Five-week-old female C57BL/6 mice were housed in a cage maintained at 23 °C in a 12 h/12 h light/dark cycle under specific pathogen-free conditions. After 1 week of adaptation, 6-week-old mice weighing 18–22 g were used for the experiments. We divided mice into five groups: Group 1 (n = 10), WT mice as vehicle control group; Group 2 (n = 10), WT mice as NSAID-treated disease group; Group 3 (n = 10), WT mice as NSAIDs-treated disease group administered with 50 mg/kg walnut-containing pellet diet for 2 weeks; Group 4 (n = 10), WT mice as NSAIDs-treated disease group administered with 100 mg/kg walnut-containing pellet diet for 2 weeks; and Group 5 (n = 10, WT mice as NSAIDs-treated disease group administered with 200 mg/kg walnut-containing pellet diet for 2 weeks. Based on our previous studies, WPE doses (50, 100, and 200 mg/kg) were determined in animal experiments [56]. Gastric ulcers were induced in mice by intra-gastric administration of IND. C57BL/6 mice were fasted for 24 h and then administered either saline or IND (50 mg/kg) by oral gavage. For gastric ulcer damage model, mice were sacrificed 16 h later and then stomach tissues were collected to analyze detailed molecular study. Animals were handled in an accredited animal facility in accordance with the Association for Assessment and Accreditation of Laboratory Animal Care International (AALAC in CHA Bio Complex of CHA University) policies. The animal study was approved by Center of Animal Care and Use (CACU) committee in CHA University (Approval number#16-0903).

### 4.6. Hematoxylin and Eosin (H&E) Staining and Immunohistochemistry

For histopathological analysis, the stomach and small intestine were fixed in 10% neutralized buffered formalin, processed using the standard method, and embedded in paraffin. Sections of 4 μm thickness were then stained with H&E. The glandular mucosae of corpus and antrum were examined histologically. The pathological changes in NSAIDs treatment, such as inflammatory cells infiltration, erosive lesions, and ulceration, were graded by three gastroenterologists, who were blinded to the group, using an index of histologic injury defined [56]. In this study, inflammation was defined as grade of the infiltration of inflammatory cells, 0: none, 1: under the lamina propria, 2: half of mucosa, and 3: until the epithelial gland layer (all mucosa). The erosion was defined as proportion of erosive lesion, 0: none, 1: loss of epithelial gland layer (1/3 proportion), 2: two-three portion of mucosa (2/3 proportion), and 3: all mucosa (3/3 proportion).

For the immunohistochemistry, paraffin unstained slides were deparaffinized, rehydrated, and boiled in 100 mM Tris-buffered saline (pH 7.6) with 5% urea in an 850 W microwave oven for 5 min each. Sections were also incubated with antibody in the presence of 0.1% bovine serum albumin and finally incubated for 24 h at 4 °C. The sections were counterstained with hematoxylin.

### 4.7. Terminal Deoxynucleotidyl Transferase-Mediated dUTP Nick End Labeling (TUNEL) Assay

To determine cytotoxicity using TUNEL method using in situ cell apoptosis detection kits (Promega, Madison, WI, USA), the paraffin block tissue slides were depaffinized and permeabilized with paraformaldehye. The slides were incubated with the UNEL reaction mixture contatiningTdT and fluorescein-dUTP or TMR-dUTP. During this incubation step, TdT catalyzed the attachment of fluorescein-dUTP to free 3′OH ends in the DNA and then visualized the incorporated fluorescein with a fluorescence microscope.

### 4.8. Cytokine Array and ELISA

Mouse cytokine antibody arrays were purchased from R&D Systems (Minneapolis, MN, USA) and carried out strictly according to manufacturer’s instruction. Three hundred micrograms of a representative case of each group were used for this assay. ELISA kits for mouse IL1β and IL-6 (R&D Systems) were purchased and used strictly according to the manufacturer’s instruction.

### 4.9. Statistical Analysis

Data from three independent experiments at least were expressed as the mean ± SD. The statistical significance of differences between two groups was evaluated using Student’s test. Analysis was performed using Sigma plot (Version 10). Statistical significance was accepted at *p* < 0.05, unless otherwise indicated.

## 5. Conclusions

In conclusion, we confirmed that the dietary intake of walnut, plentiful in essential fatty acids, vitamins, and polyphenols, prevented NSAID-induced gastric damage. This is an extended study of a previous publication that WPEs attenuated H. pylori-associated inflammation [55,56]. In the current study, we documented that a dietary intake of walnut could attenuate the gastric inflammation and protect the gastric mucosa via the concerted actions of the attenuated inflammatory mediators, increased preservation of 15-PGDH, and enhanced gastric defense mechanisms. Walnut seems to be very effective and attractive as a well-acknowledged “pharmanutrient” against NSAID-induced GI damage. However, a well-designed clinical trial should follow this study.

## Figures and Tables

**Figure 1 ijms-25-07239-f001:**
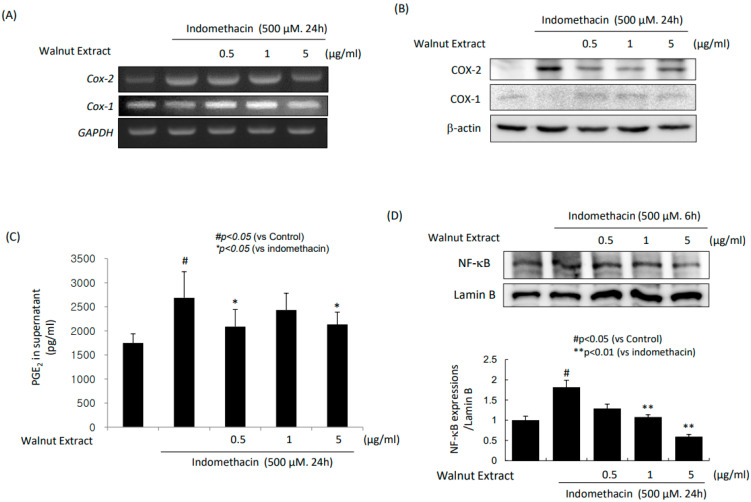
Changes in COXs, PGE2, and NF-κB after indomethacin alone and combination of indomethacin and WPE. (**A**) RT-PCR for *COX-1* and *COX-2* mRNA. (**B**) Western blot for COX-1 and COX-2. (**C**) ELISA for PGE2. (**D**) Western blot for NF-κB in nuclear fraction. The data represent mean ± SD (n = 3). Significant differences between the compared groups are indicated (^#^
*p* < 0.05 vs. control, * *p* < 0.05 vs. indomethacin, and ** *p* < 0.01 vs. indomethacin).

**Figure 2 ijms-25-07239-f002:**
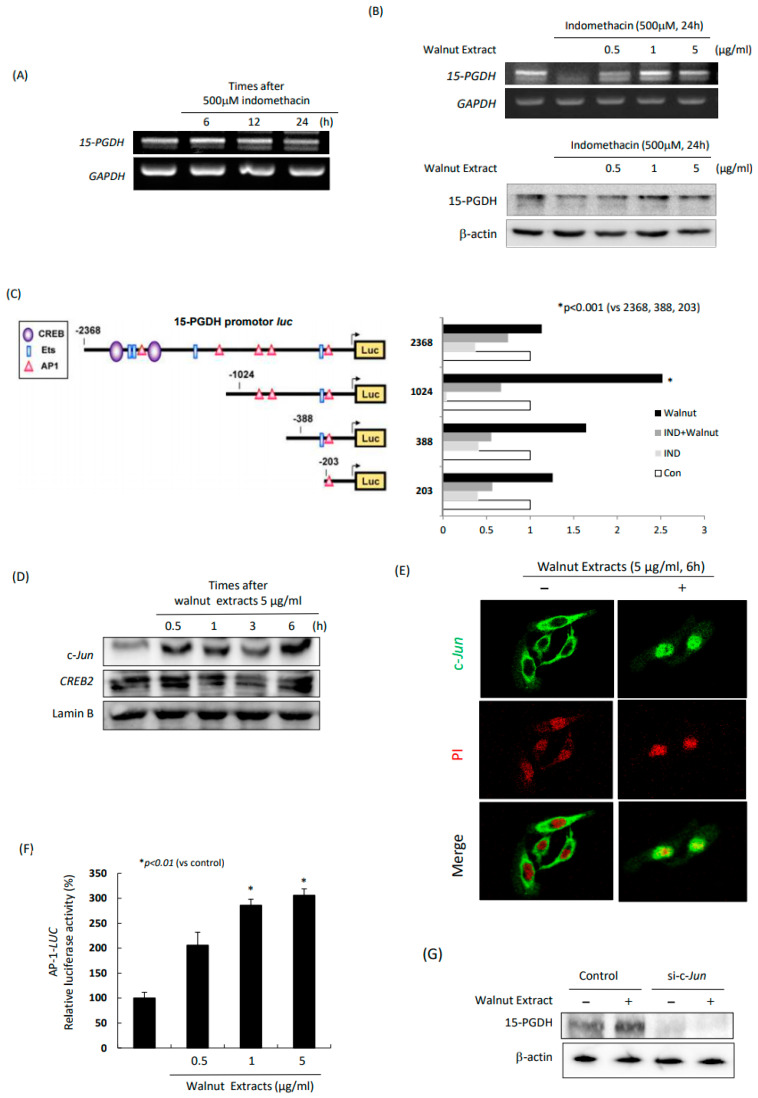
Changes in 5-PGDH after indomethacin alone and combination of indomethacin and WPE. (**A**) RT-PCR for *15-PGDH* mRNA. (**B**) 15-PGDH after indomethacin alone and combination of indomethacin and WPE; RTPCR (upper) and Western blot (lower). (**C**) Promoter assay of 15-PGDH, −2368, −1024, −388, and −203 constructs was generated as shown in left figure. Promoter activities were measured after WPE, indomethacin (IND) and WPE, indomethacin alone (IND), and control. (**D**) Western blot for c-Jun, CREB2 in nuclear extracts after indomethacin alone, and combination of indomethacin and WPE. (**E**) Confocal imaging of c-Jun and PI antibody staining; merged figures were presented. (**F**) AP-1 luciferase activity measurement after different doses of WPEs (0.5, 1, and 5 μg/mL of WPE, respectively). (**G**) Western blot for 15-PGDH after WPE administration in Mock-transfected and sic-Jun-transfected cell homogenates.

**Figure 3 ijms-25-07239-f003:**
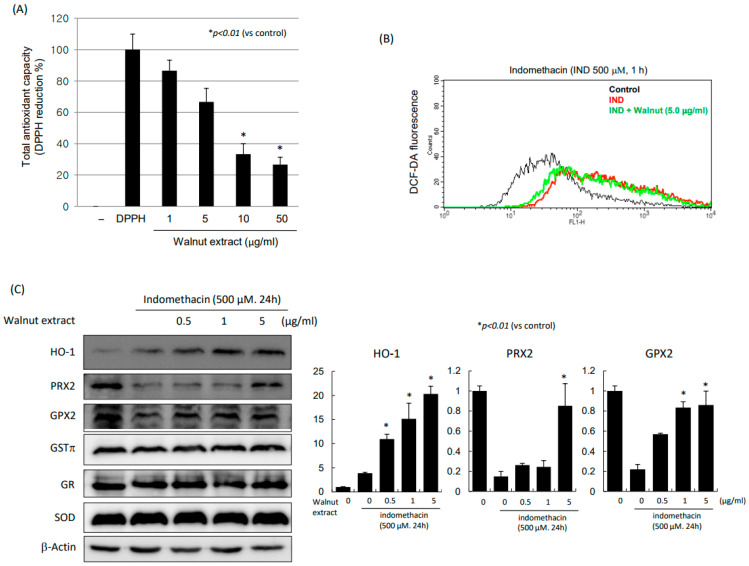
Changes in oxidative stress and antioxidative enzymes after indomethacin alone and combination of indomethacin and WPE. (**A**) DPPH reduction assay after indomethacin alone and combination of indomethacin and different doses of WPEs (1, 2.5, 5, 10, 25, 50, and 100 μg/mL of WPE). (**B**) DCF-DA assay by flow cytometry showing generation of DCF fluorescence. (**C**) Western blot for antioxidative enzymes. (**D**) RT-PCR for *NOX-1* mRNA. (**E**) RT-PCR for *HO-1* and *HSPs* mRNA. (**F**) HO-1 luciferase activities after different doses of WPEs (0.5, 1, and 5 μg/mL).

**Figure 4 ijms-25-07239-f004:**
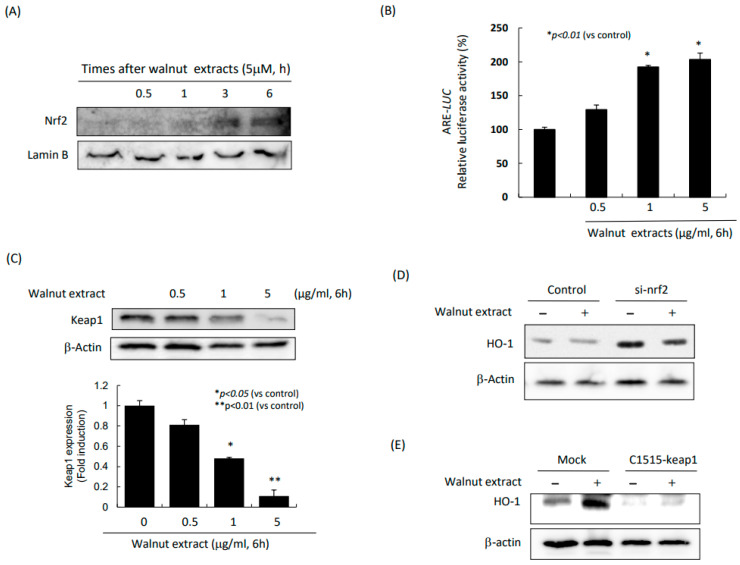
Changes in Nrf2 and HO-1 after indomethacin alone and combination of indomethacin and WPE. (**A**) Western blot for Nrf2. (**B**) ARE (antioxidant response element)-luciferase activities after WPE. (**C**) Western blot for keap1. (**D**) Western blot for HO-1 after WPE administration in Mock- and siNrf2-transfected cells. (**E**) Western blot for HO-1 after WPE administration in control- and C1515-keap1 inhibitor-treated cells.

**Figure 5 ijms-25-07239-f005:**
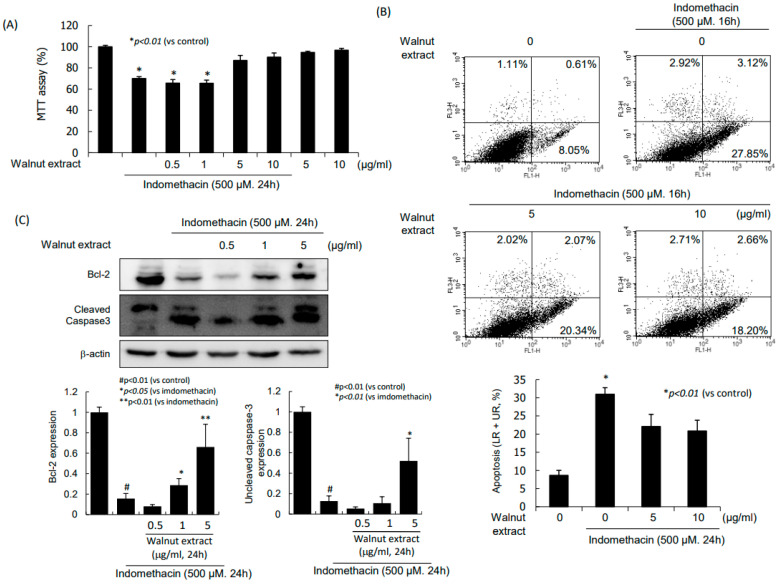
Inhibitory action of NSAID-induced apoptosis with WPE. (**A**) Cell viability assessed with MTT assay. (**B**) Flow cytometry to measure apoptotic cells. (**C**) Western blot for bcl-2 and caspase-3 after indomethacin alone and combination of indomethacin and WPE.

**Figure 6 ijms-25-07239-f006:**
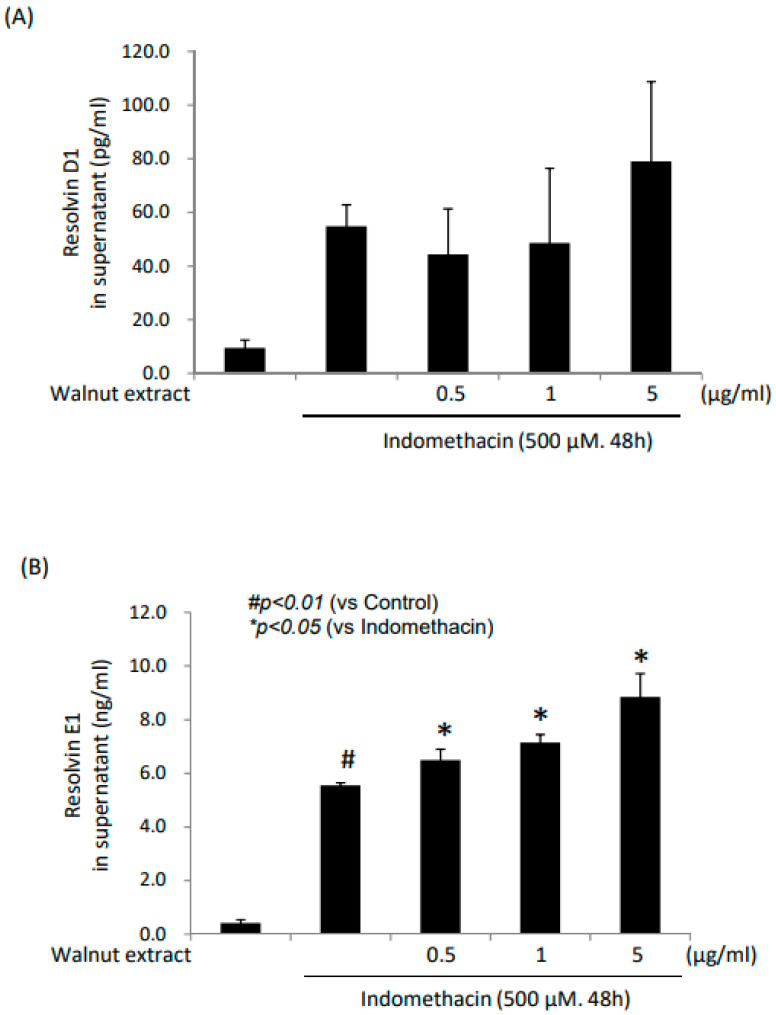
Levels of resolvin D1 (RvD1) and resolvin E1 (RvE1). (**A**) RevD1; (**B**) RevE1.

**Figure 7 ijms-25-07239-f007:**
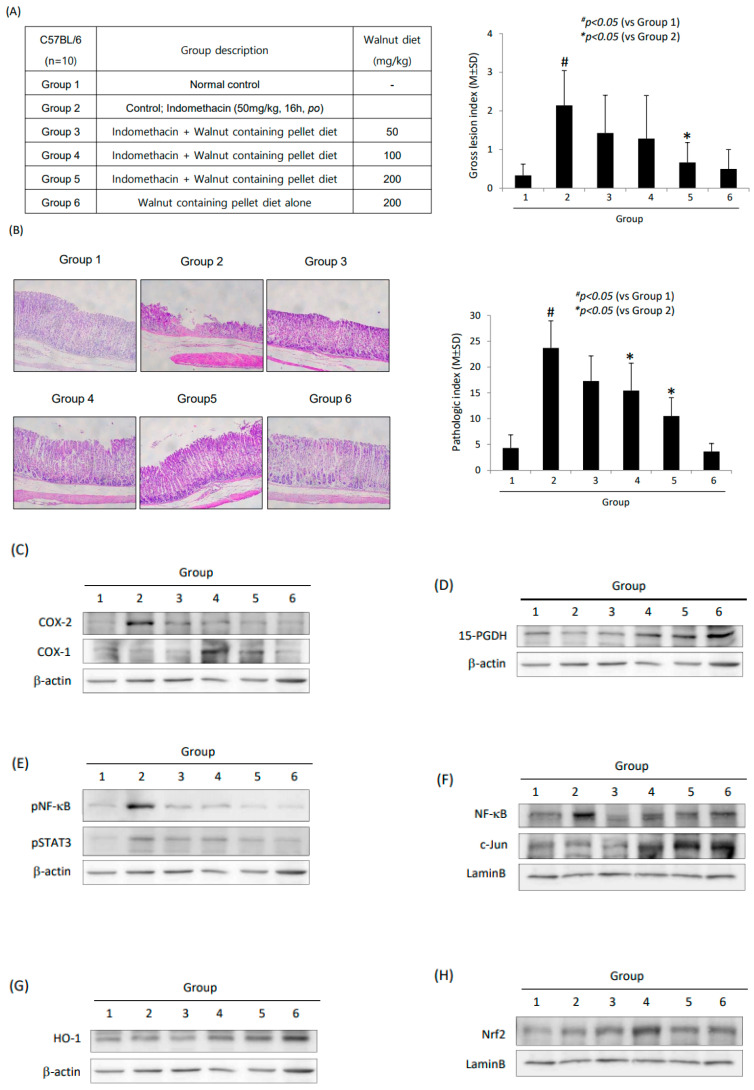
Indomethacin-induced gastric damage model; effect of walnut-containing diet on mitigating indomethacin-induced gastric damage. (**A**) Schemes for groups: Group 2 is indomethacin-induced gastric damage, Group 3 is indomethacin-induced gastric damage treated with 50 mg/kg walnut-containing diet, Group 4 is indomethacin-induced gastric damage treated with 100 mg/kg walnut-containing diet, Group 5 is indomethacin induced gastric damage treated with 200 mg/kg walnut-containing diet, and Group 6 is rats administered 2000 mg/kg walnut-containing diet only. Gross lesion index was shown according to group. (**B**) Representational pathological photo according to group. In Group 2, our model showed significant development of erosion, shallow ulcers, and inflammatory cell infiltrations. Mean pathological scores were presented. (**C**) Western blot for COX-1 and COX-2 in gastric mucosal homogenates according to group. (**D**) Western blot for COX-1 and COX-2 in gastric mucosal homogenates according to group. (**E**) Western blot for 15-PGDH in gastric mucosal homogenates according to group. (**F**) Western blot for p-NF-κB and p-STAT3 in gastric mucosal homogenates according to group. (**F**) Western blot for NF-κB and c-*Jun* in nuclear fraction of gastric mucosal homogenates according to group. (**G**) Western blot for HO-1 in gastric mucosal homogenates according to group. (**H**) Western blot for Nrf2 in nuclear fraction of gastric mucosal homogenates according to group.

**Figure 8 ijms-25-07239-f008:**
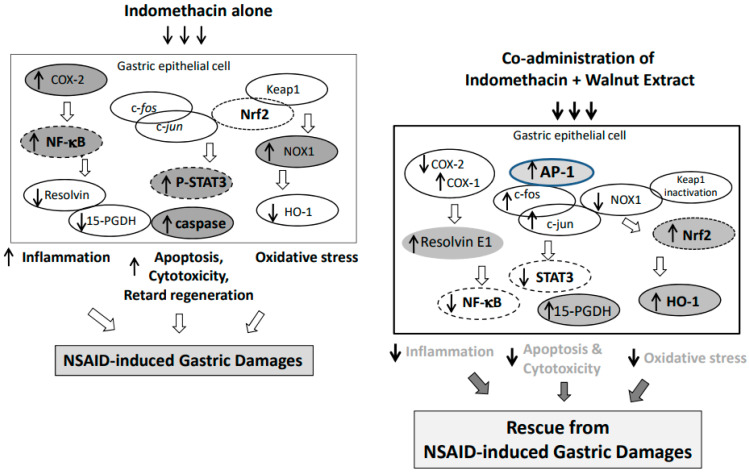
Schematic drawing explaining the molecular pathogenesis of indomethacin-induced gastric damage; increased COX-2, increased NF-κB, increased NOX-1, increased STAT3, and increased caspase-3, whereas decreased 15-PGDH, decreased HO-1, decreased RVE1, and retarded the activation of c-*Jun*/c-*Fos* relevant to repressed transcriptional activation of gastroprotective 15-PGDH, leading to significant development of indomethacin-induced gastric damage. However, co-administration of indomethacin and walnut significantly reversed these molecular pathogenic events, rescuing from indomethacin-induced gastric damage.

## Data Availability

Data is contained within the article and Appendix A.

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
