# Peer review of "Dietary Walnuts Prevented Indomethacin-Induced Gastric Damage via AP-1 Transcribed 15-PGDH, Nrf2-Mediated HO-1, and n-3 PUFA-Derived Resolvin E1"

_ijms, 2024, doi:10.3390/ijms25137239_

Round 1

Reviewer 1 Report

Comments and Suggestions for Authors

This study on the preventive effects of walnut extract (WPE) against indomethacin-induced gastric mucosal inflammation is highly intriguing and well-written. It will undoubtedly inspire other researchers in this field. However, I kindly suggest addressing the following points to further enhance this work:

  1. The authors have used the rat gastric mucosal cell line RGM1 in this experiments. It would be beneficial to provide brief additional data demonstrating similar activities of WPE in human gastric epithelial cell lines such as AGS and MKN-45. This would strengthen the translational relevance of the findings.
  2. Please elaborate on the rationale behind the dosage selection for WPE in animal experiments. For instance, mention whether author conducted preliminary studies or relied on literature review to determine the appropriate doses (50, 100, and 200 mg/kg). 
  3. The WPE extract likely contains various bioactive compounds, but there are no direct results on these specific components in this mauscript. In the discussion section, please explore the potential key active ingredients responsible for the observed effects. Moreover, discuss future research directions, such as investigating the underlying mechanisms of action for these compounds.

Author Response

Thank you for valuable comments and please find the response in a-point-to-point manner.

Reviewer 2 Report

Comments and Suggestions for Authors

El study “Dietary walnuts prevented indomethacin-induced gastric damages via AP-1 transcribed 15-PGDH, Nrf2-mediated HO-1, and n-3 PUFA-derived resolvin E1” is interesting and provides evidence of the efficacy and mechanisms of walnut extract against indomethacin-induced gastric damage. However, some important points need to be addressed:

1.    In figures 1C and 1D, there are symbols representing differences between the groups; however, it is not mentioned in the figure legend which groups are being compared.

2.    Please, review the statistical analysis carefully, as the data variability suggests that the groups in figure 1C may not be significantly different.

3.    In the figure description, please review the spelling of NF-KB, as there appears to be an error.

4.    In the description of figure 3, there appears to be an error in the units of the WPE treatment.

5.    Provide a label for the y-axis in figure 4C.

6. The bar graphs shown in figure 7 do not describe which groups are being compared and which groups show differences. As with figure 1, the variability is very high, suggesting that the groups may not be significantly different. Please verify the statistical analysis.

7.    In the methodology, the authors should describe the treatment protocols for both in vitro and in vivo experiments, including the doses of indomethacin and walnut extract, treatment/exposure times for each, how long before the indomethacin was administered, and justify these choices.

8.    Mention the source of the walnut extract.

9.    In section 4.1, explain what EPA is.

10. In section 4.5, it is described that 4 groups were created, but in the results and figures, there are more than 4 groups. It seems that some parts of the methodology correspond to another study, as a fat-1 TG group is described, which was not used in this work.

11. Describe the number of rodents in each group and how many repetitions were performed for the in vitro treatments.

12. It is mentioned that the rodents were sacrificed at 16 and 48 hours of treatment; however, these results are not shown.

13. In section 4.6, it is mentioned that the damage was quantified/analyzed using a pathological index. Describe what this index consists of, the criteria for defining the grades, and how many grades this index includes. It is also mentioned that the analysis was carried out by 3 specialists. What was done with these analyses? Was an average of the results from the 3 specialists taken?

14. Why is quantification of the Western blot results performed for some cases but not for others? That is, some are accompanied by bar graphs and others are not. Please ensure consistency by showing bar graphs for all.

15. Please, Add a conclusion to your work.

Author Response

(The authors gave the same response as above.)

Round 2

Reviewer 2 Report

Comments and Suggestions for Authors

The authors have responded to the comments, I have no more comments and I consider that the manuscript has been improved.